**Brief Communication**

# GPU-accelerated homology search with MMseqs2

**Felix Kallenborn** [1,10]**, Alejandro Chacon**[2,10]**, Christian Hundt**[2]**, Hassan Sirelkhatim** [2]**, Kieran Didi**[2,3]**, Sooyoung Cha** [4,5]**, Christian Dallago** [2,6,7] ✉**, Milot Mirdita** [4] ✉**, Bertil Schmidt** [1] ✉ **& Martin Steinegger** [4,5,8,9] ✉

Rapidly growing protein databases demand faster sensitive search tools. Here the graphics processing unit (GPU)-accelerated MMseqs2 delivers 6× faster single-protein searches than CPU methods on 2 × 64 cores, speeds previously requiring large protein batches. For larger query batches, it is the most cost-effective solution, outperforming the fastest alternative method by 2.4-fold with eight GPUs. It accelerates protein structure prediction with ColabFold 31.8× over the standard AlphaFold2 pipeline and protein structure search with Foldseek by 4–27×. MMseqs2-GPU is available under an open-source license at https://mmseqs.com/.

Many biological advances rely on computationally identifying evolutionarily related sequences (homologs)[1–4] from vast reference databases to infer protein properties[5–7]. Remote homologs are pivotal to deep learning methods[8–10] to predict accurate three-dimensional (3D) structures[11–13].

The Smith–Waterman–Gotoh algorithm[14,15] guarantees an optimal gapped alignment through dynamic programming[16]; however, it is impractically slow for large sequence databases[12]. Therefore, heuristic methods such as BLAST[1], PSI-BLAST[17], MMseqs2 (ref. 4) and DIAMOND[3] use seed-and-extend filters to prune dissimilar sequences before costly gapped alignment. In contrast, sensitive aligners like HMMER[2] and HHblits[18] utilize dynamic programming-based filters to compute the highest-scoring gapless matches.

To accelerate both heuristic and sensitive approaches, several optimizations have been explored, including CPU-specific instruction sets, parallelization, and specialized hardware such as field-programmable gate arrays[19] and GPUs[20,21].

Here, we introduce two GPU-accelerated algorithms (Fig. 1a,b) integrated into MMseqs2 for gapless filtering and gapped alignment using position-specific scoring matrices (PSSMs)[22]. These achieve speedups over CPU-based methods without sacrificing sensitivity, greatly accelerating homology retrieval, query-centered multiple

sequence alignment (MSA) generation for structure prediction and structural searches with Foldseek[23].

The GPU-accelerated gapless filter maps query PSSMs to columns and reference sequences to rows in a matrix, processing each matrix row in parallel, while utilizing shared GPU memory to optimize access to PSSMs (Fig. 1c) and packed 16-bit floating-point numbers to maximize throughput. CUDASW++4.0 (ref. 24) was incorporated and modified to operate on PSSMs, using a wavefront pattern to handle dynamic programming dependencies.

On random amino acid sequences, the gapless GPU kernel achieved speedups of up to 2.8× on one L40S GPU (peak 13.5 trillions of cell updates per second or TCUPS) and 21.4× on eight L40S GPUs (peak 102 TCUPS) relative to a 2 × 64-core CPU server (Fig. 1d), outperforming previous acceleration methods by one-to-two orders of magnitude[19,20]. On real protein sequences, speedups increased to 18.4× for one L40S and 110× for eight L40S, while TCUPS peaked at 11.3 and 67.5 TCUPS, respectively (Fig. 1e, Methods and Supplementary Data 1). These results surpass prior accelerated gapless filter methods, which reached 1.7 TCUPS on an Alveo U50 field-programmable gate array[19] and 0.4 TCUPS on a K40 GPU[20].

For fair comparison, we parameterized all homology tools to match ROC1 sensitivity of ~0.40. Iterative profile searches, where initial

[1]Department of Computer Science, Johannes Gutenberg University Mainz, Mainz, Germany. [2]NVIDIA, Santa Clara, CA, USA. [3]Department of Computer Science, University of Oxford, Oxford, UK. [4]School of Biological Sciences, Seoul National University, Seoul, Republic of Korea. [5]Interdisciplinary Program in Bioinformatics, Seoul National University, Seoul, Republic of Korea. [6]Department of Biostatistics and Bioinformatics, Duke University, Durham, NC, USA. [7]Department of Cell Biology, Duke University, Durham, NC, USA. [8]Institute of Molecular Biology and Genetics, Seoul National University, Seoul, Republic of Korea. [9]Artificial Intelligence Institute, Seoul National University, Seoul, Republic of Korea. [10]These authors contributed equally: Felix Kallenborn, Alejandro Chacon. ✉e-mail: cdallago@nvidia.com; mmirdit@snu.ac.kr; bertil.schmidt@uni-mainz.de; martin.steinegger@snu.ac.kr

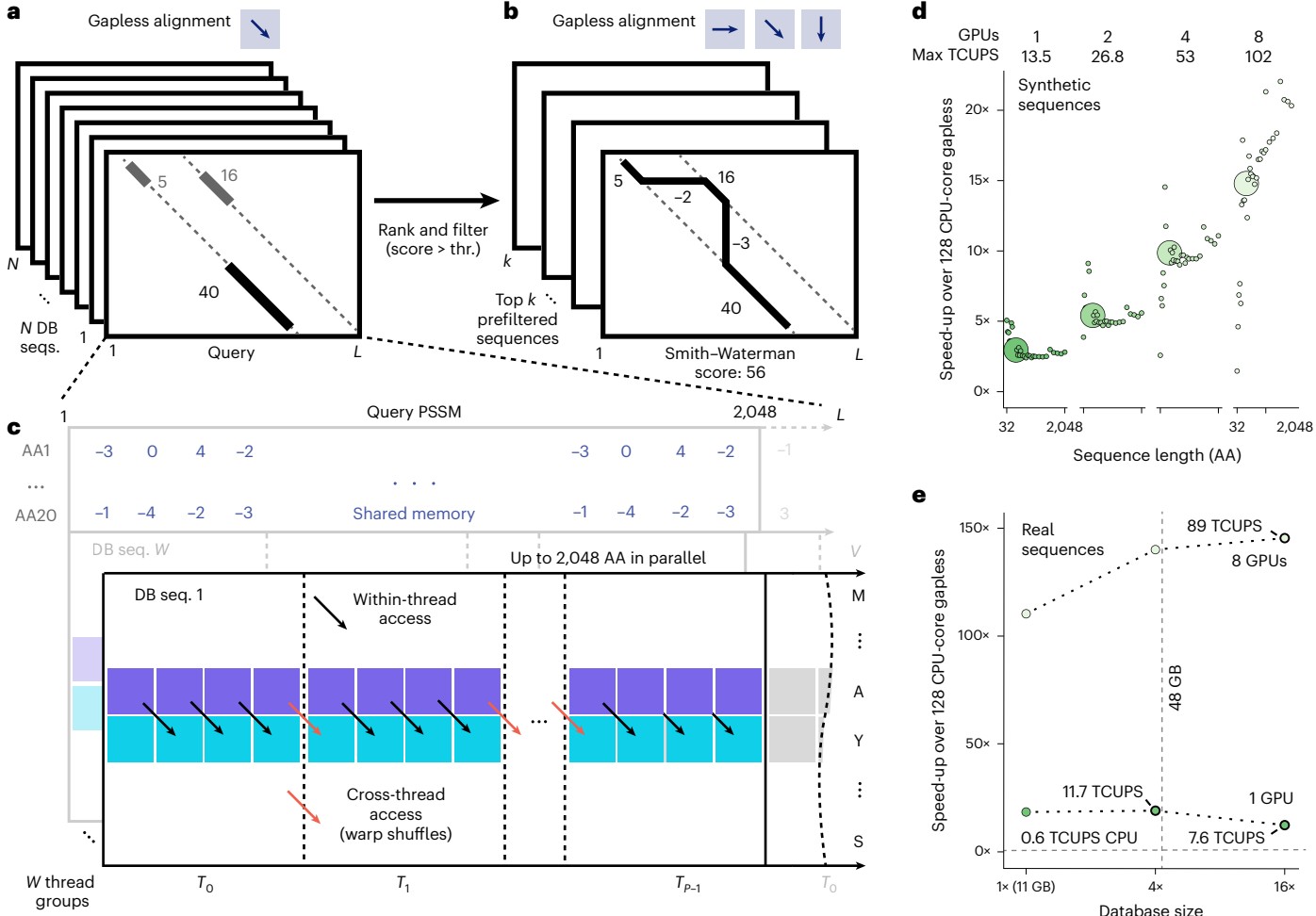

**Fig. 1 | MMseqs2-GPU workflow and gapless alignment performance.**
**a**, Gapless alignment scans reference sequences against a query, ranking and filtering them by alignment scores. **b**, Sequences above a threshold proceed to gapped Smith–Waterman–Gotoh alignment. **c**, GPU-optimized gapless alignment splits the query profile into segments (up to 2,048 residues), loading them into fast shared memory for efficient access by GPU threads; warp shuffles allow efficient cross-thread data sharing for diagonal computations.

**d**, GPU speedups (1, 2, 4 and 8 L40S GPUs) relative to a 2 × 64-core CPU for random sequence pairs (lengths 32–2048). **e**, GPU speedups (1 and 8 GPUs) versus a 2 × 64-core CPU for 6,370 queries searching against a 1×, 4× and 16× sized 30-million-protein reference database. The 16× set exceeds GPU memory, requiring database streaming at 7.575/11.676 TCUPS ≈ 64.9% of in-memory performance.

hits are converted into PSSMs for subsequent search rounds, further improve sensitivity, with MMseqs2-GPU achieving 0.612 and 0.669 ROC1 after two and three iterations, surpassing PSI-BLAST (0.591) and approaching JackHMMER (0.685; Methods and Supplementary Data 1).

We benchmarked homology search speed in a single query, and a large batch scenario of 6,370 queries, both against a ~30-million-sequence database (Fig. 2a). MMseqs2-GPU on one NVIDIA L40S GPU outperformed JackHMMER by 177× for single queries and 199× for large batches and was 6.4× faster than BLAST (Fig. 2a) in single-query searches. For large batch sizes, MMseqs2-CPU *k*-mer on 2 × 64 CPU cores performed 2.2-fold faster, but MMseqs2-GPU became 2.4-fold faster on eight GPUs (Fig. 2a). Performance on H100 GPUs closely mirrored L40S GPUs, while A100 GPUs were 1.6× slower (Supplementary Data 1).

Although DIAMOND benefits from larger query batches, even at 100,000 queries its per-query speed plateaued at 0.42 s, while MMseqs2-GPU and MMseqs2-CPU *k*-mers achieved faster speeds of 0.37 s and 0.17 s per query, respectively (Methods and Supplementary Data 1).

For cloud cost estimates, we compared AWS EC2 pricing. MMseqs2-GPU on a single L40S instance was the least-expensive

option across all batch sizes. MMseqs2-CPU *k*-mer on 2 × 64 cores was 60.9× and 1.6× more costly for single-batch and large-batch workloads, respectively (Fig. 2a).

We measured energy consumption in the single-batch scenario, where MMseqs2-GPU achieved highest efficiency with four L40S GPUs, showing 80.7-fold and 2.1-fold better efficiency compared to JackHM-MER and MMseqs2-CPU *k*-mer, respectively. A lower-cost configuration with 16 CPU cores and one L4 GPU yielded a 95-fold improvement over JackHMMER and a 2.5-fold improvement over MMseqs2-CPU (Supplementary Data 1).

MMseqs2's *k*-mer-based filtering was optimized over time for small batch sizes on web servers[25] and rapid MSA generation in ColabFold[26], but required substantial RAM (up to 2 TB). MMseqs2-GPU reduces this memory demand from ~7 bytes to 1 byte per residue, supports further reduction via clustered searches, and allows distributing databases across multiple GPUs or streaming from host RAM at 63–65% of in-GPU-memory speed (Fig. 1e, Extended Data Fig. 1 and Methods).

GPU acceleration also benefits structure prediction pipelines. While alternative methods trade accuracy for higher throughput[27], query-centered MSA methods remain the most accurate[11]. We compared end-to-end structure prediction pipelines: AlphaFold2

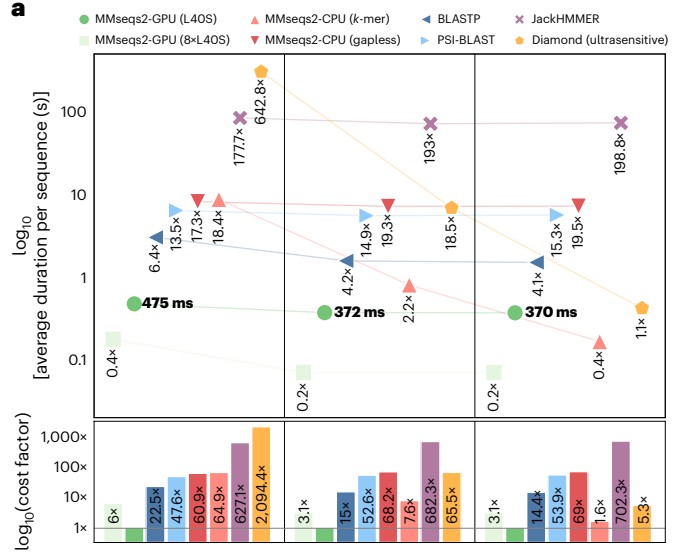

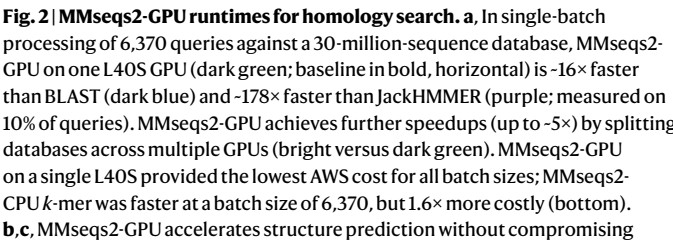

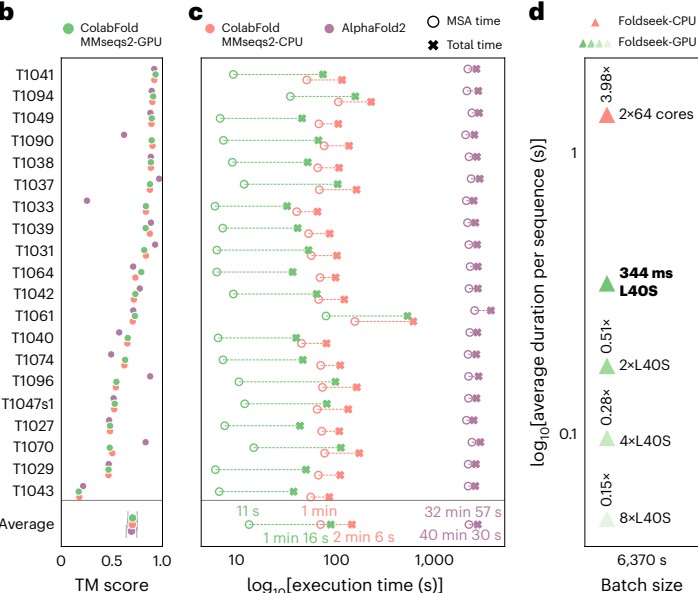

**Fig. 2 | MMseqs2-GPU runtimes for homology search. a**, In single-batch processing of 6,370 queries against a 30-million-sequence database, MMseqs2-GPU on one L40S GPU (dark green; baseline in bold, horizontal) is ~16× faster than BLAST (dark blue) and ~178× faster than JackHMMER (purple; measured on 10% of queries). MMseqs2-GPU achieves further speedups (up to ~5×) by splitting databases across multiple GPUs (bright versus dark green). MMseqs2-GPU on a single L40S provided the lowest AWS cost for all batch sizes; MMseqs2-CPU *k*-mer was faster at a batch size of 6,370, but 1.6× more costly (bottom). **b,c**, MMseqs2-GPU accelerates structure prediction without compromising accuracy (0.70 ± 0.05 TM-score). On 20 CASP14 targets, ColabFold MMseqs2-GPU (green) was 1.65× faster than ColabFold-CPU *k*-mer (orange) and 31.8× faster than AlphaFold2 (JackHMMER+HHblits, violet). MMseqs2 searched 238 million cluster representatives and expanded to 1 billion members; JackHMMER searched 426 million sequences, and HHblits searched 81 million profiles containing 2.1 billion members. **d**, Foldseek-GPU on one L40S (dark green, baseline in bold, horizontal) is 4× faster than Foldseek-CPU *k*-mer (orange) at large batch sizes (6,370 queries). Eight L40S GPUs accelerate searches by 7× compared to one GPU, and 27× compared to Foldseek-CPU.

(JackHMMER+HHblits), ColabFold MMseqs2-CPU *k*-mer and ColabFold MMseqs2-GPU, on 20 CASP14 free-modeling targets[28]. AlphaFold2 searches a database of 506 million entries, with 426 million sequences and 81 million profiles (representing ~2.1 billion members). Both Colab-Fold pipelines initially perform two rounds of three-iteration searches against 238 million cluster representatives before expanding and realigning to roughly one billion sequences (Methods), highlighting its suitability for metagenomics-scale searches.

ColabFold with MMseqs2-GPU was 1.65× faster than MMseqs2-CPU and 31.8× than AlphaFold2, driven primarily by accelerated MSA generation (5.4× and 176.3× faster, respectively). AlphaFold2's CPU-based MSA step consumes 83% of runtime, MMseqs2-GPU cuts this to 14.7%, enabling integrated single-GPU execution (Fig. 2c and Supplementary Data 1). All methods achieved similar accuracy (0.70 ± 0.05 Template modeling score (TM-score); Fig. 2b and Supplementary Data 1).

MMseqs2's modular architecture serves as the backbone for many methods, for example, the protein structure aligner Foldseek[23], whose high sensitivity requirement reduces the advantage of *k*-mer-based searches. We benchmarked Foldseek's search speed for CPU and GPU using 6,370 structures sampled from AlphaFold Database clusters (50% sequence identity). Foldseek-GPU on one L40S GPU was 4× faster than Foldseek-CPU *k*-mer (2 × 64 CPU cores) and 27.3× faster on eight GPUs (Fig. 2d), while modestly improving sensitivity in a SCOPe benchmark: family (0.874 versus 0.861), superfamily (0.493 versus 0.487) and fold recognition (0.108 versus 0.106).

MMseqs2-GPU ranks as the fastest and cheapest evolutionary search tool across various experiments. Unlike word-based search methods[1,3,4], MMseqs2-GPU's sensitive gapless filter, does not allow trading sensitivity for speed. However, reaching high sensitivity is essential for many use-cases, such as structure prediction[11].

Our CPU-based benchmarks ran on a server with 1 TB RAM and 256 threads, resources typically inaccessible to many researchers.

MMseqs2-GPU addresses this limitation by lowering memory requirements and offloading computations onto GPUs. On Google Colab Pro using a cost-effective NVIDIA L4 GPU (24 GB RAM) with a modest CPU (six cores, 64 GB RAM), MMseqs2-GPU delivers a 10-fold speedup over JackHMMER for searching UniRef90 (2022_01, 144 million proteins; Methods and Supplementary Data 1). Potential GPU memory constraints are mitigated through reduced memory footprint, efficient database streaming, partitioning and clustered searches.

MMseqs2-GPU accelerates protein homology search and structure prediction without sacrificing accuracy, improving throughput at lower cost (Fig. 2). Many bioinformatics tools, such as orthology inference[29,30], structure alignment[23] and GPU-powered workflows limited by homology retrieval speed (for example, retrieval-augmented protein language models[31]), can directly benefit. Thus, MMseqs2-GPU broadens access to rapid, cost-effective homology searches.

## Online content

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

## Methods

### Background: fast sequence comparison principles

Filtering is an essential step in homology search to reduce the overall execution time by ranking all reference database sequences quickly and filtering a subset against which the computationally more demanding Smith–Waterman–Gotoh algorithm can be run. As one of the earliest tools, BLAST[1] proposed index structures for initial seed finding with bidirectional gapless extension to match a given minimum score.

MMseqs2 (ref. 4) introduced a filtering approach based on double-consecutive $k$-mer matches on the same diagonal. Through this approach, all $k$-mers of a reference database are stored in a random-access memory (RAM)-based index structure for quick retrieval (hereafter, 'the index'). For single-query searches at high sensitivity, MMseqs2 generates long lists of similar $k$-mers from the query sequence and matches them against the index to check the filtering criterion of two consecutive matches on one diagonal. While highly optimized, random accesses to the index results in poor cache locality.

Diamond[3] generates lists of $k$-mers for both queries and references to sort and compare them colinearly. Instead of similar $k$-mers, Diamond utilizes multiple spaced-$k$-mer patterns. Colinear comparison results in improved cache locality at the expense of indexing and sorting overhead.

In contrast to word-based filtering approaches, HMMER[2] and HHblits[18] implemented a more sensitive, albeit slower ranking technique by simplifying the Smith–Waterman–Gotoh algorithm to perform a 'gapless' alignment (that is, excluding gaps). In essence, this allows finding the longest common subsequence consisting of residue substitutions only between any pair of sequences, allowing for better resolution than word-based methods.

Modern hardware accelerators like GPUs lend themselves to highly parallel workflows through their high core count, albeit at lower operational complexity per core. Thus, the gapless filtering is well suited to exploit their capabilities, due to reduced instruction count with little data dependencies, while additionally avoiding branching and random memory accesses typically used in $k$-mer index lookups.

### MMseqs2-GPU algorithm and parallelization

Pairwise gapless alignments are computed between a query represented by the PSSM $Q$ and each reference sequence. $Q$ is constructed either through a single-query sequence and a substitution matrix, or from a sequence profile, based on previous search results. Once computed, $Q$ provides a score for placing any amino acid at any position along the length of the query. Next, our 'gapless filter' computes a pairwise local alignment between the reference sequence $S = (s_1, \ldots, s_n)$ and the PSSM $Q$ of length $m$ by dynamic programming, and populates a matrix $M$ using the recurrence relation $M[i,j] = \max\{M(i-1, j-1) + Q[i, s_j], 0\}$ for all $1 \le i \le m, 1 \le j \le n$. Initialization is given by $M[i, 0] = M[0, j] = 0$. Algorithm 1 outputs the maximum value in $M$, which represents the score of the optimal local alignment without gaps (also known as gapless) between $S$ and $Q$. The top $k$ sequences passing an inclusion threshold are then passed to a Smith–Waterman–Gotoh algorithm operating on profiles. Both of these algorithmic steps are executed on a GPU.

**Algorithm 1.** *Gapless filter score computation*
**Input:** Protein sequence $S[]$, of size $n$, PSSM $Q[]$ of size $m \times |\Sigma|$
**Output:** Score
Score $\leftarrow 0$
**for** $I \leftarrow 0$ **to** $m$ **do**
$\quad M[i, 0] \leftarrow 0$
**end**
**for** $j \leftarrow 0$ **to** $n$ **do**
$\quad M[0, j] \leftarrow 0$
**end**
**for** $i \leftarrow 1$ **to** $m$ **do**
$\quad$ **for** $j \leftarrow 1$ **to** $n$ **do**
$\quad\quad M[i, j] \leftarrow \max(M[i-1, j-1] + Q[i, s_j], 0)$
$\quad\quad$ Score $\leftarrow \max($Score, $M[i, j])$
$\quad$ **end**
**end**

### CPU-SIMD gapless filter

MMseqs2 offers a gapless filter accelerated on CPUs through single-instruction multiple-data (SIMD) instructions, through the `ungappedprefilter` module first introduced in MMseqs2 6.f5a1c. In this work, we describe the module, its integration into the MMseqs2 `search` workflow with `--prefilter-mode 1` and its extension[32] to incorporate soft-masking of low-complexity regions with tantan[32]—in addition to the GPU integration described below.

The filtering process begins by preparing a striped query profile[33] on a single thread, and finally utilizes all available CPU cores to linearly compare all reference sequences on disk. During filtering, all soft-masked target database amino acids are represented as the unknown X residue.

At its core, the gapless alignment follows Farrar's[33] striped Smith–Waterman–Gotoh algorithm first used in HMMER[2], which is adjusted to compute only the gapless diagonal. It avoids affine gap computation and requires only five SIMD operations to update a striped query segment to retrieve the maximum gapless diagonal score.

The algorithm uses 8-bit integers to represent alignment scores to maximize the parallel SIMD register usage. Scores that exceed the 8-bit limit of 255 are clamped to 255, which indicates a strong potential match. Such matches are then aligned using the full Smith–Waterman–Gotoh algorithm. Depending on the supported instruction set of the CPU, this gapless implementation uses 128-bit (SSE2) or 256-bit (AVX2) vector size. We also evaluated AVX512 and found only a marginal performance benefit and did not implement it. In addition, our implementation exploits multi-core CPUs by computing different target sequences independently using multi-threading.

### GPU-accelerated gapless filter

**Core algorithm.** We present a new implementation of a gapless filter on GPUs, designed to leverage the simplified dependency scheme inherent in gapless alignments for high parallelism and performance. Unlike the Smith–Waterman–Gotoh algorithm, where each cell in the dynamic programming matrix depends on its top, left and diagonal neighbors, gapless alignment reduces this dependency to only the diagonal neighbor. This crucial simplification enables parallel computation across all cells within the same row, potentially enhancing processing speed in comparison to more complicated parallelization schemes, such as wavefront parallelization typically used to accelerate gapped Smith–Waterman computation. However, to realize this potential, the design of an appropriate parallelization scheme is required that optimizes necessary memory accesses and inter-thread communication while efficiently handling highly variable sequence lengths.

**Memory accesses and data types.** To optimize memory accesses and performance, the query PSSM is stored in shared memory, facilitating fast, multi-threaded access (Fig. 1d). Meanwhile, reference sequence residues, which map to the $y$ axis of the dynamic programming matrix, are stored in global memory, requiring one byte per residue. Our calculations use 16-bit floating-point numbers or 16-bit integers, supported by the DPX instructions available with NVIDIA's Hopper GPU architecture, packed into a 32-bit word using half2 or s16x2 data types.

**Thread grouping and tile optimization.** Drawing inspiration from CUDASW++4.0 (ref. 24), our implementation assigns each gapless alignment task to a thread group, with typical sizes of 4, 8 or 16 threads. This configuration allows multiple thread groups to handle different alignments concurrently. The dynamic programming matrix is

processed in a row-by-row manner, with each row partitioned among threads that are responsible for up to 128 cells each. To optimize performance across various query lengths, we use different tile sizes, which are determined by the product of the thread group size and the number of columns processed per thread. They are realized as template parameters, with the group size constrained to be a divisor of 32, and half the number of columns being a multiple of 4. Optimal tile size configurations for different query lengths were identified via a separate grid search program, minimizing out-of-bounds computations when the query length is less than the tile size. As the processing begins, reference sequence residues are loaded from global memory, and each thread accesses its corresponding values from the scoring profile. To boost throughput, we designed the memory access pattern so that each thread loads eight consecutive 16-bit scores from shared memory in a single instruction.

**Shared memory bank conflict mitigation.** We further optimized shared memory access for thread groups of size four to mitigate bank conflicts. Shared memory is organized into 32 four-byte banks, and we utilize two-byte values in the PSSM, meaning two columns are packed per four-byte word. The PSSM is arranged in shared memory such that the $i$-th four-byte column maps to memory bank '$i \bmod 32$'.

A warp-wide load from shared memory is broken up into one or more hardware transactions of size 128 bytes served by the 32 memory banks. Bank conflicts occur when multiple accesses within the same transaction target the same bank but different addresses, leading to serialization. This can arise when multiple thread groups within a warp access the same PSSM columns but different rows (corresponding to different residues in the reference sequence). For group sizes of eight or larger, this is not an issue, as each group loads $8 \times 16 = 128$ bytes, thus fitting a single transaction. However, with a group size of four, each group loads only $4 \times 16 = 64$ bytes. This allows a second group to load from a different PSSM row (but the same columns) within the same transaction, resulting in a two-way bank conflict. To resolve this, we use two copies of the PSSM in shared memory. These copies are stored such that the $i$-th thread group consistently uses memory banks 0–15 if $i$ is even, and banks 16–31 if $i$ is odd. This ensures that accesses from different groups within the same transaction are directed to distinct memory banks, eliminating the bank conflict.

**Per-cell computation and warp shuffles.** Once the scores are loaded, threads perform computations for their assigned columns. Each thread performs the following operations for each cell: (1) adds the score from the scoring profile to the value from the previous diagonal cell, (2) sets the result to zero if it is negative, and (3) updates the local maximum score if the current cell's value exceeds it. To facilitate data communication between threads within a group, particularly for diagonal dependencies, we use warp-shuffle operations. Because the group size is at most 32, all threads within a group belong to the same warp. Warp shuffles allow for efficient register-based data exchange between threads, avoiding slower shared memory access with explicit synchronization, or the even slower use of global memory.

**Data permutation and vectorization.** To double computational throughput, we utilize hardware instructions capable of processing two 16-bit values packed into a 32-bit word. However, direct packing of neighboring matrix columns into a single 32-bit word introduces dependencies that hinder independent processing. To address this challenge, we apply a data permutation technique that rearranges columns within a thread, aligning diagonal dependencies directly within the packed data. For example, in cases where a thread handles 32 columns, we pack columns (0, 16), (1, 17), (2, 18), and so forth. This arrangement ensures that each pair of packed columns corresponds to cells that are diagonally adjacent in the dynamic programming matrix.

As a result, the dependency between packed values is restricted to the preceding value from the previous row, enabling full exploitation of vectorized operations for computational efficiency.

**Long sequence handling.** This strategy allows matrix tiles with up to 2,048 columns to be processed efficiently with a thread group of size 16. For protein sequences exceeding 2,048 residues, the profile matrix is divided into multiple tiles processed sequentially. In such cases, the last column of each tile is stored in global memory to serve as the starting point for the first column of the next tile.

**Multi-GPU parallelization.** To maximize performance on systems with multiple GPUs, we distribute the reference database across the available GPUs. Specifically, the target database sequences are split into chunks, and each GPU processes a separate chunk against the query sequence. The results from each GPU are then combined to produce the final result list. This parallelization strategy reduces execution time and allows for processing larger databases by leveraging the combined available GPU memory.

**Database streaming.** We implemented several optimizations to increase processing speed and handle large datasets efficiently. We minimize host–GPU communication latency by partitioning the reference database into smaller batches, allowing processing to be pipelined via asynchronous CUDA streams. Here, data transfer of the next batch $i + 1$ from the host to the GPU is overlapped with the processing of the current batch $i$ on GPU. Notably, this approach allows us to efficiently handle databases larger than the available GPU memory.

Furthermore, to maximize GPU memory utilization and minimize data transfer, the workflow caches as many reference database batches as can fit in GPU memory, and reuses them for subsequent queries without retransfer from the host. The streaming capability further extends to reading database batches directly from disk storage when the dataset exceeds available host RAM.

**GPU server.** MMseqs2 workflows are constructed through scripts that repeatedly invoke the `mmseqs` binary, each time specifying different modules to execute. We observed that each invocation requiring GPU resources incurs a CUDA initialization overhead of approximately 300 ms. This startup cost can become substantial in complex workflows; for instance, during the ColabFold MSA search, the `ungapped-prefilter` module is called six times (three iterative searches each against the UniRef30 and ColabFoldDB databases).

To circumvent this overhead, we introduced an optional, dedicated GPU server mode. Here, we launch a persistent background process that maintains the GPU context and becomes responsible for database caching and executes alignment computations. Subsequent `mmseqs ungappedprefilter` invocations communicate their requests to this running server process via Linux shared memory, thereby avoiding the initialization penalty and reducing overall workflow execution time.

## GPU-accelerated Smith–Waterman–Gotoh

In addition to the gapless filter on GPU, we implemented a version of Smith–Waterman–Gotoh with affine gap penalties operating on protein profiles (PSSMs) as a modification of CUDASW++4.0 (ref. [24]) to align reference sequences to the same query profiles used in the filter. As described in the previous section, this required transposing the computed dynamic programming matrix to place the profile along the $x$ axis, and leveraged the same parallelization strategies. Matrix tiles are processed by thread groups of size 4, 8, 16 or 32 using in-register computations with 32-bit capacity to avoid overflows.

In contrast to the filter algorithm, the Smith–Waterman–Gotoh algorithm does not allow for threads in the same group to operate on the same row in parallel because cells depend on their left neighbor.

Consequently, a wavefront pattern is used to have threads work on different rows along the minor diagonals.

## Cell updates per second

The speed of dynamic programming algorithms is typically reported by converting runtime into the number of dynamic programming matrix cell updates that are performed per second; for example, TCUPS as $TCUPS = (\sum_i m_i \times n_i)/(t \times 10^{12})$, where $t$ is the runtime in seconds and $m_i$ and $n_i$ are the lengths of the aligned sequences.

**Efficiency analysis.** We analyzed whether the proposed algorithm is able to effectively remove overheads incurred by memory accesses using half2 arithmetic by modeling the theoretical peak performance (TPP) of the utilized GPU hardware as:

$$TPP = \frac{\#SMs \times Throughput\_per\_instruction \times Clock}{Cycles\_per\_cell\_update}$$

Where

- `Throughput_per_instruction` refers to the number of results per clock cycle per streaming multiprocessor (SM) of native arithmetic instructions on the considered hardware. A corresponding table for devices of various compute capabilities is provided in the CUDA documentation (https://docs.nvidia.com/cuda/cuda-c-programming-guide/index.html#arithmetic-instructions; accessed 12 May 2025).
- `Cycles_per_cell_update` models the maximum attainable performance constrained by the algorithm structure and the specifications of the architecture. In our case, referring to the theoretical minimal number of clock cycles needed by an individual SM of the utilized GPU to calculate one dynamic programming matrix cell in $M$.

`Cycles_per_cell_update` can initially be determined by the inner loop in algorithm 1 and thus set to three (that is, two `max` instructions and one `add` instruction). However, as the L40S GPU used in most of our experiments enables dual ports and floating-point operations, allowing to simultaneously issue `add` and `max` operations in a single SM cycle, `Cycles_per_cell_update` can be set to two (when using single-precision arithmetic) and to one (when using half2 arithmetic).

Disregarding the lookup operation to the PSSM, any value to register movements such as warp shuffles, any data transfers, and based on the specified `Throughput_per_instruction` of 64 for `max` instructions (the dependency bottleneck) on the L40S (compute capability 8.9), we can use the following equation to calculate the `TPP` for half2 arithmetic on the L40S as follows:

$$TPP(L40S) = \frac{142 \times 64 \times 2.56GHz}{1} = 23.2TCUPS$$

Provided the synthetic benchmark on a single L40S achieves a performance of up to 13.5 TCUPS (Fig. 1d), our approach is able to achieve an efficiency of 58% on an L40S GPU architecture, which shows that our optimizations, such as warp shuffles and PSSM lookups from shared memory, are able to effectively transform the problem to a compute-bound one and minimize overheads from memory accesses.

## MMseqs2-GPU workflow

The MMseqs2-GPU workflow starts with query and reference sets residing in CPU RAM. For the first step, that is, filtering, query profiles are transferred to the GPU and permuted to enable efficient CUDA shared memory accesses during lookup. Additionally, the reference set is partitioned into smaller batches, which are transferred to the GPU. Throughout the filtering step, gapless alignment scores are stored for subsequent sorting. In the next stage, after all reference database batches have been processed, 'gapped', that is, Smith–Waterman–Gotoh alignment scores are computed for top reference sequences

satisfying inclusion thresholds using the same partitioned approach as that used for gapless alignment. Finally, the filtered reference sequences and their corresponding alignment scores are transferred back to the CPU.

## Hardware setup

With the exception of benchmarks executed on Google Colab, the same base system hardware setup was used. Specifically, the system featured two AMD EPYC 7742 64-core (2.25 GHz; thermal design power (TDP) of 225 W) CPUs (effectively 128 physical cores running 256 logical threads and a TDP of 450 W), with 1 TB of DDR4 RAM and 2 TB NVMe Intel SSDPE storage.

For the GPU benchmarks, the base system configuration additionally included an L4, A100, L40S or H100 PCIe NVIDIA GPU, which are set at a TDP of 72, 300, 350 and 350 W, respectively. The energy efficiency measurements additionally include a one-socket EPYC 7313P 16-core (3 GHz; 155 W TDP) system, with 256 GB DDR4 RAM and L4 GPU.

## Benchmarks

**Synthetic TCUPS benchmark.** To measure TCUPS performance for MMseqs2-GPU (Fig. 1d), we performed a synthetic benchmark generating sequences of equal lengths for several lengths. For each possible dynamic programming matrix tile size $l$, a randomly generated query of length $l$ was aligned to a database of 5 million randomly generated sequences of length $l$. Runtime was then converted to TCUPS. We executed the same benchmark using the CPU implementation of the gapless alignment described in 'CPU-SIMD gapless filter'. We show the speedup for each length $l$ of the respective execution on 1, 2, 4 and 8 GPUs versus the CPU execution. This synthetic benchmark explores the best-case scenario by using uniform length, as this avoids reduced hardware utilization caused by adjacent CUDA thread groups processing different reference sequence lengths. For real-world database searches, the variable length effect is minimized by sorting the database sequences by length in ascending order.

**Database scaling benchmark.** To investigate performance characteristics of the MMseqs2-GPU `ungappedprefilter` implementation when the available GPU memory is exceeded, we measure TCUPs on real amino acid sequences based on the sequence sets described in 'Sensitivity', with additional measurements conducted by extending the reference sequences 4, and 16 times (Fig. 1e). We extrapolated the runtime for the 16× replicated database on one L40S from a subset of 500 random queries of all 6,370 queries. TCUPS are similar for MMseqs2-GPU combined gapless and gapped executions for 1, 2, 4, 8 and 16 times the reference sequence set (Extended Data Fig. 1).

**Sensitivity.** The same approach and datasets described in MMseqs2 (ref. 4) were used to conduct the sensitivity benchmark. This benchmark involved annotating full-length UniProt sequences with structural domain annotations from SCOP[34], designating 6,370 sequences as queries and 3.4 million as reference sequences. The full-length query sequences included disordered, low-complexity and repeat regions, which are known to cause false-positive matches, especially in iterative profile searches. Additionally, 27 million reversed UniProt sequences were included as reference sequences (resulting in a total of 30,430,281 reference sequences).

In line with previous work, true-positive matches are defined as those with annotated SCOP domains from the same family, while false positives match reversed sequences or sequences with SCOP domains from different folds. The sensitivity of a single search is measured by the area under the curve before the first false-positive match (ROC1), indicating the fraction of true-positive matches found with a better $E$-value than the first false-positive match. Sensitivity results for the various tools and modes, measured as ROC1, are summarized in the table below, with full benchmark details available in Supplementary Data 1.

| Tool | Mode | Sens. 1-it | 2-it | 3-it |
|------|------|-----------|------|------|
| MMseqs2-CPU | `-s 8.5` | 0.391 | 0.606 | 0.665 |
| MMseqs2-CPU | Gapless | 0.4 | 0.612 | 0.669 |
| MMseqs2-GPU | Gapless | 0.4 | 0.612 | 0.669 |
| BLAST | | 0.401 | | |
| PSI-BLAST | | | 0.574 | 0.591 |
| JackHMMER | | 0.4 | 0.614 | 0.685 |
| Diamond | Ultra sens | 0.409 | | |

**Speed.** For speed benchmarks, the same query and reference sequences as those used for the sensitivity benchmarks were used, and methods' parameters were set to reach comparable sensitivity where possible. Specifically, we ran the following methods:

- **JackHMMER (v3.4)**[2]: at one iteration (equivalent to a phmmer search), with an *E*-value cutoff of 10,000, using 128 threads (corresponding to the real number of cores, which differs from the actual number of threads), omitting the alignment section for the generated output.
- **BLAST (v2.16.0)**[1]: with an *E*-value cutoff of 10,000, using 128 threads and limiting the number of filtered reference sequences to 4,000.
- **PSI-BLAST (v2.16.0)**[1]: at two iterations, with an *E*-value cutoff of 10,000, using 128 threads and limiting the number of filtered reference sequences to 4,000.
- **Diamond ultra sensitive (v2.1.9)**[3]: using `blastp`, the ultra-sensitive setting, with an E-value cutoff of 10,000, using 128 threads and limiting the number of filtered reference sequences to 4,000.
- **MMseqs2-CPU *k*-mer commit (16-747c6)**[4]: using `easy-search` and the sensitivity at 8.5, with an *E*-value cutoff of 10,000, using 128 threads and limiting the number of filtered reference sequences to 4,000.
- **MMseqs2-CPU gapless commit (16-747c6)**[4]: using `easy-search` and the prefilter mode at 1, with an *E*-value cutoff of 10,000, using 128 threads and limiting the number of filtered reference sequences to 4,000.
- **MMseqs2-GPU commit (16-747c6)**: using `easy-search`, the prefilter mode at 1 and enabling the GPU usage via the `--gpu 1--gpu-server 1` options, with an *E*-value cutoff of 10,000, using one thread, and limiting the number of filtered reference sequences to 4,000.

We measured the total execution time of these methods based on the invocations above and derived an average per-sequence execution. Additionally, we set up searches in either single-batch mode, emulating the behavior in, for example, a protein 3D structure prediction server, or by batching sequences into groups of 10 and 100 sequences, or one batch of 6,370 sequences, emulating the annotation of small protein sets or proteomes.

Due to the prohibitive execution time of JackHMMER[2] in any batch mode, or Diamond[3] in single-batch mode, we reduced the query set to only 10%, or 637 query sequences to extrapolate results for the remaining cases. As batch 6370 would require us to run 100% of the query set, and batching results from 1 to 100 for 10% of the query set showed a negligible effect for JackHMMER, we excluded running this final benchmark with sustainability in mind.

**Diamond scaling.** Diamond[3] was developed to annotate vast metagenomic databases, and its performance is tuned to large query sets and batch sizes. To allow a fair comparison of Diamond's performance, we thus performed a query database and batch scaling benchmark exclusively for Diamond. For this benchmark, all previous execution parameters were retained with the exception of the query database, because

upsampling the 6,370-query set to obtain larger query database sizes is not a realistic setting. Instead, we randomly sampled (without replacement) from UniRef90 (ref. 35) to obtain query databases of size 100, 1,000, 10,000 and 100,000 sequences. We set the batch size equal to the database size for each run.

**Structure prediction.** To compare total runtime and the effect that MSA choice has on structure prediction accuracy, we replicated the ColabFold[26] benchmarks to predict the CASP14 (ref. 28) free-modeling queries.

We measured MSA input generation and model inference times for the three configurations listed below. For all three configurations, we omit template search and relaxation and execute the default three recycling iterations.

- **AlphaFold2**: a baseline-configuration using the AlphaFold2 (ref. 8) pipeline, utilizing JackHMMER[2] and HHblits[18] to compute input MSAs from three databases, and structure prediction inference using AlphaFold2 weights. Specifically, this configuration utilizes the following homology search databases: UniRef90 2020_01 (JackHMMER, 139 million sequences), MGnify 2018_12 (JackHMMER, 287 million sequences), Uniclust30 2018_08 (HHblits, 15 million profiles and 124 million in total), BFD first release (HHblits, 66 million profiles and 2.1 billion in total). Five AlphaFold models were run on one L40S GPU via the docker container from https://github.com/google-deepmind/alphafold/ (commit f251de6).
- **ColabFold MMseqs2-CPU**: a configuration using ColabFold v1.5.5, utilizing MMseqs2 (commit 22115b) with a default *k*-mer on one CPU to compute input MSAs, and structure prediction inference loading AlphaFold2 weights. By default, ColabFold utilizes recent versions of the databases; however, we retrieved older versions to approximate a fair comparison to the AlphaFold2 configuration, specifically UniRef30 2021_03 (29 million representatives and 277 million in total) and ColabFoldDB 2021_08 (209 million representatives and 739 million in total). Five AlphaFold models were run on one L40S GPU using ColabFold code obtained from https://github.com/sokrypton/ColabFold/ (commit 09993a8).
- **ColabFold MMseqs2-GPU**: a configuration using ColabFold v1.5.5, utilizing MMseqs2-GPU (commit 22115b) with default gapless running on one L40S GPU to compute input MSAs, and structure prediction inference loading AlphaFold2 weights. We used the same databases and codebase as in the ColabFold MMseqs2-CPU configuration. All five models were run on one L40S GPU.

Structure prediction inference in all three configurations leveraged an L40S. To compare structure prediction accuracy, we computed several metrics between the predicted and ground-truth structures, and focused results on template modeling score (TM-score[36]).

**Cloud cost.** We compared cloud cost to run the homology search methods presented in the speed benchmark. To do so, we retrieved the on-demand hourly cost for AWS EC2 instances (retrieved on 24 September 2024) that best matched our hardware setup for instances in the US East (Ohio) region. We obtained:

- **CPU-based methods**: for JackHMMER, Diamond (ultra sensitive), MMseqs2-CPU *k*-mer and BLAST, we selected the instance 'c7a.32xlarge' with 128 physical CPU cores and 256 GB of system RAM at an hourly cost of US$6.57.
- **GPU-based method**: for MMseqs2-GPU, we selected the instance 'g6e.xlarge' with 2 physical CPU cores and 32 GB of system RAM and one L40S GPU at an hourly cost of US$1.86.

For cloud costs utilizing 2 and 4 L40S, we selected the instance 'g6e.12xlarge' with 24 physical CPU cores and 384 GB of system RAM and 4 L40S GPUs at an hourly cost of US$10.493. For cloud cost calculations utilizing 8 L40S, we selected the instance 'g6e.48xlarge' with 96 physical CPU cores and 1,536 GB of system RAM and 8 L40S GPUs at an hourly cost of $30.131.

To obtain the per-query cost, we multiplied the hourly cost by the actual total runtime for each method and divided by the number of queries. All results can be found in Supplementary Data 1 ('Cloud cost estimates'). Note that prices of cloud instances can differ regionally, and over time. To attempt a robust comparison, we reported the cost factor (that is, the difference in cost compared to a baseline, which we selected to be MMseqs2-CPU $k$-mer) rather than the total theoretical cost.

**Energy consumption.** To measure average power utilization, we leveraged `powerstat` and `nvidia-smi`, which allow sampling hardware counters (running average power limit) included in CPUs and GPUs, and their memory. We performed this measurement for the full query set in single-batch mode for various methods. We then multiplied the power average by the execution times to obtain the total energy consumption.

**Foldseek.** We benchmarked Foldseek[23] using MMseqs2-GPU by sampling 6,370 protein structures represented as 3D interaction (3Di) strings (average length: 261 3Di letters). These were sampled from AFDB50 v4 (53.6 million entries, average length: 264 3Di letters) against the same database. The database was indexed using `createindex` and stored in memory.

- **Foldseek-CPU k-mer commit c438b9 (ref. [23]):** using `search` with the option `--db-load-mode 2` for fast index reading and 128 threads `--threads 128`.
- **Foldseek-GPU commit c438b9:** using `search` with the option `--db-load-mode 2` for fast index reading, 128 threads `--threads 128` and `--gpu 1` for gapless GPU alignment.

We retrieved the Foldseek SCOPe-based sensitivity benchmark from https://github.com/steineggerlab/foldseek-analysis/ (commit 1737c71). Foldseek-GPU was executed with parameters `--max-seqs 2000 -e 10 --gpu 1`. Foldseek-CPU $k$-mer was executed with parameters `--s 9.5 --max-seqs 2000 -e 10`.

**Colab benchmark.** To compare the speed of MMseqs2-GPU to JackHMMER on a more typically encountered hardware setup, we chose a paid Google Colab Pro environment with a 6-core/12-thread CPU, 64 GB of system RAM and a NVIDIA L4 GPU with 24 GB RAM. We searched the same 20 CASP14 sequences as described in 'Structure prediction' against the UniRef90 2022_01 (containing 144 million proteins; benchmark performed on 6 October 2024). As the reference database is 48.6 GB, it does not fully fit into GPU memory (24 GB), leveraging system RAM streaming. We chose the following parameters for the search:

- **JackHMMER v3.4:** at one iteration (equivalent to a phmmer search), with an $E$-value cutoff of 10,000, using 12 threads and omitting the alignment section for the generated output.
- **MMseqs2-GPU commit 81ddab:** using `easy-search`, enabling the GPU usage via the `--gpu 1` options, with an $E$-value cutoff of 10,000, using 12 threads and limiting the number of filtered reference sequences to 4,000.

### Reporting summary

Further information on research design is available in the Nature Portfolio Reporting Summary linked to this article.

### Data availability

Data utilized to perform benchmarks for this study are freely available. For speed, sensitivity and energy consumption benchmarks, we leveraged target sequences stored at mmseqs.steineggerlab.workers.dev/targetdb.fasta.gz and reference sequences stored at mmseqs.steineggerlab.workers.dev/query.fasta. For the folding benchmark, CASP14 targets are available at predictioncenter.org/casp14/index.cgi, while the reference ColabFold databases are available at https://colabfold.mmseqs.com/. For Foldseek benchmarks, we retrieved data and scripts from https://github.com/steineggerlab/foldseek-analysis/.

### Code availability

All code developed in this study is available under MIT license and documented at https://mmseqs.com/. Analysis scripts are available at https://github.com/steineggerlab/mmseqs2-gpu-analysis/. MMseqs2 version 747c64c (as used in the paper) was deposited on Zenodo via https://doi.org/10.5281/zenodo.14223631 (ref. [37]).

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

### Acknowledgements

We thank Y. Peng, M. Stadler and M. Baust for invaluable suggestions and support, A. Elofsson and S. A. Shah for reporting issues and providing feedback, E. L. Karin, and the anonymous RECOMB-seq reviewers for critical feedback on the manuscript. M.S. acknowledges support by the National Research Foundation of Korea grants (2020M3A9G7103933, RS-2021-NR061659, RS-2021-NR056571, RS-2024-00396026, RS-2024-00438101), Samsung DS Research Fund, Creative-Pioneering Researchers Program and Novo Nordisk Foundation (NNF24SA0092560). M.M. acknowledges support from the National Research Foundation of Korea (grant RS-2023-00250470). B.S. acknowledges support from the Deutsche Forschungsgemeinschaft (DFG, German Research Foundation) – project number 439669440 TRR319 RMaP TP C01. The funding body did not participate in the design of the study and collection, in analysis and interpretation of data and in writing the manuscript.

### Author contributions

F.K., software, writing—original draft and writing—review and editing. A.C., software, investigation, writing—original draft and writing—review and editing. C.H., resources and project administration. H.S., visualization, investigation and validation. K.D., software and investigation. S.C., software and investigation. C.D., conceptualization, resources, writing—original draft, writing—review and editing, supervision, visualization and project administration. M.M., conceptualization, methodology, software, investigation, writing—original draft, writing—review and editing and supervision. B.S., software, methodology, conceptualization, resources, writing—original draft, writing—review and editing and supervision. M.S.,

software, methodology, conceptualization, resources, writing—original draft, writing—review and editing and supervision.

**Funding**

**Competing interests**

C.D., A.C., C.H., H.S. and K.D. are employed by NVIDIA. M.S. declares an outside interest in Stylus Medicine. The other authors declare no competing interests.

**Additional information**

**Extended data** is available for this paper at https://doi.org/10.1038/s41592-025-02819-8.

**Correspondence and requests for materials** should be addressed to Christian Dallago, Milot Mirdita, Bertil Schmidt or Martin Steinegger.

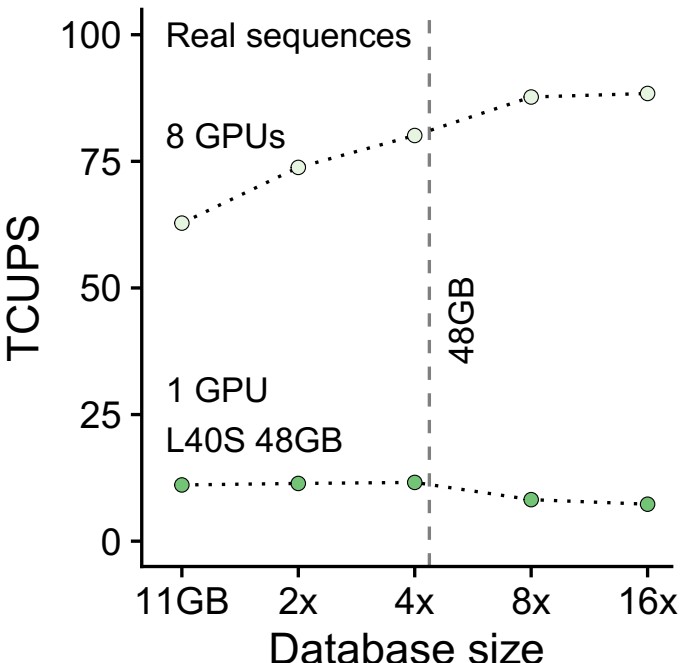

**Extended Data Fig. 1 | Combined gapless and gapped alignment TCUPS.** TCUPS of 1 and 8 GPU executions of the combined MMseqs2-GPU gapless and gapped alignment workflow for 6370 queries against target sets of 1, 2, 4, 8, and 16 times a 30 M protein database (Methods 'Sensitivity'). 8 and 16 times executions exceed GPU RAM and are processed with database streaming. The latter is processed with $7.3/11.6 TCUPS \approx 63\%$ of in-memory processing speed.

# Reporting Summary

## Statistics

For all statistical analyses, confirm that the following items are present in the figure legend, table legend, main text, or Methods section.

| n/a | Confirmed | |
|---|---|---|
| ☐ | ☒ | The exact sample size (*n*) for each experimental group/condition, given as a discrete number and unit of measurement |
| ☐ | ☒ | A statement on whether measurements were taken from distinct samples or whether the same sample was measured repeatedly |
| ☒ | ☐ | The statistical test(s) used AND whether they are one- or two-sided<br>*Only common tests should be described solely by name; describe more complex techniques in the Methods section.* |
| ☒ | ☐ | A description of all covariates tested |
| ☒ | ☐ | A description of any assumptions or corrections, such as tests of normality and adjustment for multiple comparisons |
| ☐ | ☒ | A full description of the statistical parameters including central tendency (e.g. means) or other basic estimates (e.g. regression coefficient) AND variation (e.g. standard deviation) or associated estimates of uncertainty (e.g. confidence intervals) |
| ☒ | ☐ | For null hypothesis testing, the test statistic (e.g. *F*, *t*, *r*) with confidence intervals, effect sizes, degrees of freedom and *P* value noted<br>*Give P values as exact values whenever suitable.* |
| ☒ | ☐ | For Bayesian analysis, information on the choice of priors and Markov chain Monte Carlo settings |
| ☒ | ☐ | For hierarchical and complex designs, identification of the appropriate level for tests and full reporting of outcomes |
| ☒ | ☐ | Estimates of effect sizes (e.g. Cohen's *d*, Pearson's *r*), indicating how they were calculated |

*Our web collection on statistics for biologists contains articles on many of the points above.*

## Software and code

Policy information about availability of computer code

| Data collection | Code to collect data was developed in-house and is available at https://github.com/steineggerlab/mmseqs2-gpu-analysis |
|---|---|
| Data analysis | Code to analyze data was developed in-house and is available at https://github.com/steineggerlab/mmseqs2-gpu-analysis |

For manuscripts utilizing custom algorithms or software that are central to the research but not yet described in published literature, software must be made available to editors and reviewers. We strongly encourage code deposition in a community repository (e.g. GitHub). See the Nature Portfolio guidelines for submitting code & software for further information.

## Data

Policy information about availability of data

All manuscripts must include a data availability statement. This statement should provide the following information, where applicable:
- Accession codes, unique identifiers, or web links for publicly available datasets
- A description of any restrictions on data availability
- For clinical datasets or third party data, please ensure that the statement adheres to our policy

Data utilized to perform benchmarks for this study are freely available. For speed, sensitivity, and energy consumption benchmarks, we leveraged target sequences stored at https://mmseqs.steineggerlab.workers.dev/targetdb.fasta.gz and reference sequences stored at https://mmseqs.steineggerlab.workers.dev/query.fasta. For the folding benchmark, CASP12 targets are available at https://predictioncenter.org/casp12/index.cgi, while the reference ColabFold databases are available at https://colabfold.mmseqs.com. For FoldSeek benchmarks, we retrieved data and scripts from https://github.com/steineggerlab/foldseek-analysis.

## Research involving human participants, their data, or biological material

Policy information about studies with human participants or human data. See also policy information about sex, gender (identity/presentation), and sexual orientation and race, ethnicity and racism.

| Reporting on sex and gender | N/A |
|---|---|
| Reporting on race, ethnicity, or other socially relevant groupings | N/A |
| Population characteristics | N/A |
| Recruitment | N/A |
| Ethics oversight | N/A |

Note that full information on the approval of the study protocol must also be provided in the manuscript.

# Field-specific reporting

Please select the one below that is the best fit for your research. If you are not sure, read the appropriate sections before making your selection.

☒ Life sciences  ☐ Behavioural & social sciences  ☐ Ecological, evolutionary & environmental sciences

For a reference copy of the document with all sections, see nature.com/documents/nr-reporting-summary-flat.pdf

# Life sciences study design

All studies must disclose on these points even when the disclosure is negative.

| Sample size | Sample sizes reflect those determined in  previous studies |
|---|---|
| Data exclusions | N/A |
| Replication | N/A |
| Randomization | N/A |
| Blinding | N/A |

# Reporting for specific materials, systems and methods

We require information from authors about some types of materials, experimental systems and methods used in many studies. Here, indicate whether each material, system or method listed is relevant to your study. If you are not sure if a list item applies to your research, read the appropriate section before selecting a response.

## Materials & experimental systems

| n/a | Involved in the study |
|---|---|
| ☒ ☐ | Antibodies |
| ☒ ☐ | Eukaryotic cell lines |
| ☒ ☐ | Palaeontology and archaeology |
| ☒ ☐ | Animals and other organisms |
| ☒ ☐ | Clinical data |
| ☒ ☐ | Dual use research of concern |
| ☒ ☐ | Plants |

## Methods

| n/a | Involved in the study |
|---|---|
| ☒ ☐ | ChIP-seq |
| ☒ ☐ | Flow cytometry |
| ☒ ☐ | MRI-based neuroimaging |

# Plants

Seed stocks

N/A

Novel plant genotypes

N/A

Authentication

N/A

