## [Peer Review File · Nature Methods]

GPU-accelerated homology search with MMseqs2

Corresponding Author: Professor Christian Dallago

Version 0:

Decision Letter:

27th Mar 2025

Dear Christian,

Your Brief Communication, "GPU-accelerated homology search with MMseqs2", has now been seen by 2 reviewers. As you will see from their comments below, although the reviewers find your work of considerable potential interest, they have raised a number of concerns. We are interested in the possibility of publishing your paper in Nature Methods, but would like to consider your response to these concerns before we reach a final decision on publication.

We therefore invite you to revise your manuscript to address these concerns. In particular, we would like to see a more extreme test case to justify scalability and speed claims like Reviewer #2 points out.

Link Redacted

We hope to receive your revised paper within 6 weeks. If you cannot send it within this time, please let us know. In this event, we will still be happy to reconsider your paper at a later date so long as nothing similar has been accepted for publication at Nature Methods or published elsewhere.

OPEN SCIENCE REQUIREMENTS

REPORTING SUMMARY AND EDITORIAL POLICY CHECKLISTS

EXTENDED DATA FIGURES

DATA AVAILABILITY

All novel DNA and RNA sequencing data, protein sequences, genetic polymorphisms, linked genotype and phenotype data, gene expression data, macromolecular structures, and proteomics data must be deposited in a publicly accessible database, and accession codes and associated hyperlinks must be provided in the "Data Availability" section.

CODE AVAILABILITY

Please include a "Code Availability" subsection in the Online Methods which details how your custom code is made available. Only in rare cases (where code is not central to the main conclusions of the paper) is the statement "available upon request" allowed (and reasons should be specified).

MATERIALS AVAILABILITY

ORCID

Nature Methods is committed to improving transparency in authorship. As part of our efforts in this direction, we are now requesting that all authors identified as 'corresponding author' on published papers create and link their Open Researcher and Contributor Identifier (ORCID) with their account on the Manuscript Tracking System (MTS), prior to acceptance. This applies to primary research papers only. ORCID helps the scientific community achieve unambiguous attribution of all scholarly contributions. You can create and link your ORCID from the home page of the MTS by clicking on 'Modify my Springer Nature account'. For more information please visit <http://www.springernature.com/orcid>.

Sincerely,
Arunima

Arunima Singh, Ph.D.
Senior Editor
Nature Methods

Reviewers' Comments:

Reviewer #1 (Remarks to the Author):

The manuscript describes a new method for sequence homology search implemented in GPU, taking advantage of the high performance provided by the parallelization capabilities of modern GPUs. The implementation is well described and there is a good analysis and discussion on performance, and even of costs (both dollar amounts and energy costs) which is a nice addition. This is definitely a very important contribution and addition to the landscape of sequence homology search tools.

I would have a few comments below.

Page 5. Section Online methods, A. I understand that the second prefiltering strategy introduced in paragraph "In contrast to word-based filtering..." is what is called "gapless" in subsequent paragraphs. Correct? If so, it would help to add that explicitly in the explanation: "... albeit slower ranking technique by simplifying the Smith-Waterman-Gotoh algorithm to exclude gaps from the alignment (this is termed "gapless" prefiltering)."

Page 5. Section Online methods, A: last paragraph. I think "to" is missing: ".... to exploit their capabilities, due to reduced..."

Fig 1D (and explanation in Online methods B.4): the CPU number (for 128/256 core/thread) is for mmseqs2 SIMD gapless prefilter algorithm? If so, that should be said in Figure and text.

Fig 1D and Online methods B.4: the TCUPS benchmark is for a synthetic dataset. Do authors have any idea on how real-life data affects this benchmark? Why is it not possible to provide benchmark numbers on real-life data (maybe that is obvious to the authors, but not to me)? Also more importantly: the abstract gives the TCUPS for 8GPUs. I think a fairer statement in abstract would be quoting the performance on 1 GPU (similarly to what the CUDASW++ paper does, which is as far as I can tell the one other study that uses the TCUPS metric).

Fig 2a: the labels next to the circles sometimes are in ms. Why not just in seconds like the rest of the labels? That's simply making things more confusing on an already very busy plot. By the way, the same happens in Fig 2F. Also something seems to be not right with the plotting in 2a: for the JackHMMER dots: the first data point reads 177.7s whilst the second reads 193.0s. However the second one appears lower in the y-axis in the plot. I think the same happens for "MMseqs2 (gapless)", so I suspect that something went wrong in this plotting. I suggest that the authors recheck it thoroughly. And possibly reconsider the whole plot style. It is quite difficult to follow. Also why are some labels horizontal and some vertical? Also the labels at bottom "Protein structure prediction" and "Small proteome annotation" are not making interpretation easier. I would describe the use cases in the manuscript but avoid having the labels in the chart.

Supplementary data files: the "Supplementary Data 1 PDF" is empty.

Sensitivity benchmark: the discussion on sensitivity is limited to 2 paragraphs:

Sensitivity of the new GPU-based gapless algorithm (Pages 1, 2), the discussion only talks about estimation of sensitivity in order to benchmark performance against other methods (which is definitely needed). The data is available only in a table in one of the tabs in the supplementary spreadsheet.

Sensitivity in the context of FoldSeek benchmarks (Page 3). A few numbers are given in the main text.

I think sensitivity analysis is essential to any new method like what's proposed in this manuscript. The analysis deserves some more discussion and possibly a figure with sensitivity benchmarks similar to that of previous mmseqs2 papers (can be in supplementary material). For instance, how about this question: is JackHMMER supposed to have essentially the same sensitivity as the GPU gapless method presented here (it looks like it does judging from the numbers, but it'd be nice to explicitly discuss whether there are/aren't conceptual differences in JackHMMER vs the GPU-gapless method introduced in this paper)

Reviewed by Jose Duarte

Reviewer #1 (Remarks on code availability):

I have briefly reviewed the software and documentation and it looks adequate. Reviewing the actual code is I think a job that would take many many hours for a professional software engineer, so rather unrealistic.

Reviewer #2 (Remarks to the Author):

This manuscript introduces MMseqs2-GPU, a GPU-optimized version of MMseqs2 designed for protein sequence homology searches. MMseqs2 is widely adopted by the research community, making an accelerated version valuable. However, several key limitations may hinder the widespread adoption of MMseqs2-GPU, as outlined below.

1. The speed benchmarks are based on a target database of 30 million sequences and 6,370 queries, which is significant but falls short of the scale of modern metagenomic datasets (e.g., billions of sequences). Not all people have access to GPU. MMseqs2 may be the prior option as it is already fast at such database scale. The manuscript does not address how MMseqs2-GPU performs in extreme scenarios, where GPU memory and I/O bottlenecks could become critical limitations.
2. Can MMseqs2-GPU be used to search established metagenomic databases, such as the ColabFold database? It seems that the authors already did such experiment but without much details. How to search the ColabFold database with MMseqs2-GPU? How to solve the issue of large memory requirement for big databases? According to the manuscript, it seems that MMseqs2-GPU is just 3 times (GPU vs CPU!) faster than MMseqs2 for searching the ColabFold database. This may reduce the chance of being published in Nature Methods, as it is one of the widely used options in protein structure prediction.
3. The caption for Figure 2 states, "MMseqs2-GPU can split the target database across GPUs to enable larger database searches." However, it is unclear how this splitting is implemented or achieved in practice.
4. The manuscript is difficult to follow at times. For instance, the term "TCUPS" appears in the Abstract without definition, which may confuse readers unfamiliar with its meaning. Additionally, the URL mmseqs.com redirects to GitHub. Why not directly provide the GitHub URL in the paper for clarity and convenience?
5. The documentation for the GPU-optimized version is not user-friendly. For example, following the provided instructions led to an error message: "Database DB.fasta needs header information." This suggests the instruction lacks clear guidance for users.

Reviewer #2 (Remarks on code availability):

see comments

Version 1:

Decision Letter:

Our ref: NMETH-BC59478A

5th Jun 2025

Dear Chris,

Thank you for submitting your revised manuscript "GPU-accelerated homology search with MMseqs2" (NMETH-BC59478A). It has now been seen by the original referees and their comments are below. The reviewers find that the paper has improved in revision, and therefore we'll be happy in principle to publish it in Nature Methods, pending minor revisions to satisfy the

referees' final requests and to comply with our editorial and formatting guidelines.

TRANSPARENT PEER REVIEW

ORCID

Sincerely,
Arunima

Arunima Singh, Ph.D.
Senior Editor
Nature Methods

Reviewer #1 (Remarks to the Author):

Many thanks to the authors for the enormous effort put in redoing figures and for reworking large sections of the manuscript text. I think overall the explanation of the methodology is now much clearer. I have a few comments on the new text and figures.

New abstract. I appreciate the clarifications from the previous one. And also the brevity and condensation of information is a positive point. However, I find a few of the statements a bit confusing:

- "It is most cost effective, including in large-batches at 0.45x MMseqs2-CPU speed (8 GPUs delivering 2.4x)" - I must admit that I can't make sense of this statement. Why "at 0.45x MMseqs2-CPU speed"?

- "It accelerates ColabFold structure prediction 31.8x compared to AlphaFold2". Does this mean "compared to AlphaFold2's Colab (that uses Hmmer/hhblits)". The statement as it is will only be understandable by experts that know how the different Colabs are implemented.

- "and Foldseek search 4-27x": perhaps this deserves a separate sentence. The way it is written now (connected to the other sentence) it seems to be talking about AlphaFold2 and Colabs still.

New Figure 1e: why is the first label of the x axis "11GB"? Is it the database size for 1x? I would rather keep consistent labels in x axis (1x, 4x, 16x) whilst pointing out in the title of x axis that it is relative to a database size of 11GB (30M proteins). Also I would remove the "L40S 48GB" label as an inset in the plot. It is confusing too. That should go in the legend.

New Figure 2a: I'd like to thank you the authors for the efforts in making the figure clearer. While I like most of it now, I must admit that I am not loving the extra "\$ cost" labels, because they are disconnected from the rest of the plot. The cost is sort of a 3rd dimension that's not shown. The issue for the reader is that there's no graphical impression of which method has better costs. The only way to understand it is to read every cost label and do a mental comparison. Some possible suggestions to improve it:

- Represent the cost with a variable size of the circle/triangle/cross markers. That's a common way to represent another data dimension in plots. Though I'm not sure if it would work out in practice for this case.
- Separate panel for costs analysis

- Cost analysis presented as a separate table instead of as an addition to the plot

I appreciate the very important clarifications on large hardware requirements (specially RAM) needed for CPU benchmarks that was introduced in Page 4 (to address reviewer's 2 concerns). Just one minor comment: I would be more explicit in this sentence (2nd paragraph page 5): "Our experiments ran on a server with 1TB ..." → "Our CPU-based benchmarks ran on a server with 1TB RAM ..."

Reviewer #2 (Remarks to the Author):

The authors did a nice job to improve the manuscript and the software.

Reviewer #2 (Remarks on code availability):

It works well now.

Version 2:

Decision Letter:

14th Aug 2025

Dear Chris,

I am pleased to inform you that your Brief Communication, "GPU-accelerated homology search with MMseqs2", has now been accepted for publication in Nature Methods. The received and accepted dates will be January 19, 2025 and August 14, 2025. This note is intended to let you know what to expect from us over the next month or so, and to let you know where to address any further questions.

Over the next few weeks, your paper will be copyedited to ensure that it conforms to Nature Methods style. Once your paper is typeset, you will receive an email with a link to choose the appropriate publishing options for your paper and our Author Services team will be in touch regarding any additional information that may be required.

Once proofs are generated, they will be sent to you electronically and you will be asked to send a corrected version within 48 hours. It is extremely important that you let us know now whether you will be difficult to contact over the next month. If this is the case, we ask that you send us the contact information (email, phone and fax) of someone who will be able to check the proofs and deal with any last-minute problems.

If, when you receive your proof, you cannot meet the deadline, please inform us at rjsproduction@springernature.com immediately.

If you are active on X or Bluesky, please e-mail me your and your coauthors' handles so that we may tag you when the paper is published.

To assist our authors in disseminating their research to the broader community, our SharedIt initiative provides you with a unique shareable link that will allow anyone (with or without a subscription) to read the published article. Recipients of the link

with a subscription will also be able to download and print the PDF. As soon as your article is published, you will receive an automated email with your shareable link.

Please note that you and your coauthors may order reprints and single copies of the issue containing your article through Springer Nature Limited's reprint website, which is located at <http://www.nature.com/reprints/author-reprints.html>. If there are any questions about reprints please send an email to author-reprints@nature.com and someone will assist you.

Best regards,
Arunima

Arunima Singh, Ph.D.
Senior Editor
Nature Methods

** Visit the Springer Nature Editorial and Publishing website at http://www.springernature.com/editorial-and-publishing-jobs?utm_source=ejP_NMeth_email&utm_medium=ejP_NMeth_email&utm_campaign=ejp_Nmeth for more information about our career opportunities. If you have any questions please click [here](mailto:editorial.publishing.jobs@springernature.com).**

Revision “GPU-Accelerated Homology Search with MMseqs2” (NMETH-BC59478)

Dear Arunima,

We thank you for the prompt handling of our manuscript. We also want to express our deep gratitude to Jose and the second anonymous reviewer whose inquiries have helped us improve our manuscript considerably.

In addition, we thank the RECOMB-seq reviewers for their comments on the manuscript, where we were honored to have been chosen to present a talk and poster within an overlay session during the RECOMB-seq satellite and a poster during the main conference.

To briefly summarize the comments of the RECOMB-seq reviewers: they echoed that methods were somewhat sparsely described. Therefore, we expanded technical and implementation details, such as our kernel tile-size optimization procedure, avoiding performance pitfalls due to hardware architecture, parallelization details and a theoretical analysis of achievable performance.

In addition, similar to reviewer #2, we also received feedback that the abstract and manuscript are at times difficult to follow. Therefore, we rewrote the abstract to focus less on technical details such as TCUPS and instead now express our speed-ups in relative terms. Additionally, we have substantially revised the manuscript and figures to improve clarity overall.

In regards to editorial feedback and reviewer comments on extreme test cases and scalability, we (1) rephrased our ColabFold benchmarks to highlight the ability of MMseqs2-GPU to perform metagenomic-scale searches (i.e., against a total of 238M sequences across two databases using three-iteration profile searches, with subsequent expansion and realignment to one billion cluster members), and (2) showed

MMseqs2-GPU's ability to accelerate searches against the UniRef90 2022_01 containing 144M sequences on more commonly encountered workstation hardware (using Google Collaboratory as a proxy), finally (3) we reported database scaling beyond GPU memory in a new panel in Figure 1e. This explored the behaviour of scaling the database against which searches are performed, including when exceeding GPU memory, demonstrating a moderate drop in performance at ~63% speed of in-GPU-memory processing.

Important to note is that we continue to improve our methods, including since our first submission, we reduced overheads in ColabFold MMseqs2-GPU leading to a reduced average runtime of 11s from previously 18s. Additionally, we fixed a mistake in our ColabFold MMseqs2-CPU k-mer invocation. We now report ~1 min average runtime, instead of 4 minutes previously, in a high system RAM setting for the MSA generation stage of ColabFold MMseqs2-CPU. We have updated the manuscript and accompanying supplementary data to reflect these improvements, and provide further detail in the reply to reviewer #2.

Please find in the following our point-by-point responses to the reviewers and the updated manuscript with text changes marked in **blue**.

Best regards,
Chris, Milot, Bertil and Martin

Reviewer #1

The manuscript describes a new method for sequence homology search implemented in GPU, taking advantage of the high performance provided by the parallelization capabilities of modern GPUs. The implementation is well described and there is a good analysis and discussion on performance, and even of costs (both dollar amounts and energy costs) which is a nice addition. This is definitely a very important contribution and addition to the landscape of sequence homology search tools. I would have a few comments below.

We thank the reviewer for their kind words.

Page 5. Section Online methods, A. I understand that the second prefiltering strategy introduced in paragraph “In contrast to word-based filtering...” is what is called “gapless” in subsequent paragraphs. Correct? If so, it would help to add that explicitly in the explanation: “... albeit slower ranking technique by simplifying the Smith-Waterman-Gotoh algorithm to exclude gaps from the alignment (this is termed “gapless” prefiltering).”

We thank the reviewer for detailed scanning of our manuscript for clarity. We have reworded this section:

... by simplifying the Smith-Waterman-Gotoh algorithm to perform a *gapless* alignment (i.e., excluding gaps).

As part of more substantial rewrites we have also replaced terms like “prefiltering” with the more easily understood “filtering”.

Page 5. Section Online methods, A: last paragraph. I think “to” is missing: “.... to exploit their capabilities, due to reduced...”

We fixed this typo.

Fig 1D (and explanation in Online methods B.4): the CPU number (for 128/256 core/thread) is for mmseqs2 SIMD gapless prefilter algorithm? If so, that should be said in Figure and text.

Thank you. We have redesigned Figure 1 (see below) and have updated the methods section “Cell updates per second”. In addition, we have changed the manuscript to consistently refer to MMseqs2-CPU (gapless or k-mer) or MMseqs2-GPU where appropriate (e.g. Figure 2 legend, Methods, etc.).

Fig 1D and Online methods B.4: the TCUPS benchmark is for a synthetic dataset. Do authors have any idea on how real-life data affects this benchmark? Why is it not possible to provide benchmark numbers on real-life data (maybe that is obvious to the authors, but not to me)?

We have reworked Figure 1d and added a new Figure 1e. Figure 1d shows as previous the synthetic kernel performance, however, now in terms of speedup over CPU execution. The new Figure 1e shows the ungapped speedup over CPU on the same real amino acid dataset as used in Figure 2 a/b, with additional data points for increasing database sizes. We especially thank the reviewer for this point as CPU performance was unrealistically high in the synthetic data. Now, when we compare speed-up of real and synthetic benchmarks, we observe a 18.4x speed-up in comparison to the 2.8x speed-up on synthetic sequences. The raw benchmark data is provided in Supplementary Data 1 “Synthetic gapless” and “Gapless peak performance”.

Also more importantly: the abstract gives the TCUPS for 8GPUs. I think a fairer statement in abstract would be quoting the performance on 1 GPU (similarly to what the CUDASW++ paper does, which is as far as I can tell the one other study that uses the TCUPS metric).

We have rewritten the abstract (see reviewer #2.4) to include performance of both one and eight GPUs.

Fig 2a: the labels next to the circles sometimes are in ms. Why not just in seconds like the rest of the labels? That’s simply making things more confusing on an already very busy plot. By the way, the same happens in Fig 2F. Also something seems to be not right with the plotting in 2a: for the JackHMMER dots: the first data point reads 177.7s whilst the second reads 193.0s. However the second one appears lower in the y-axis in the plot. I think the same happens for “MMseqs2 (gapless)”, so I suspect that something went wrong in this plotting. I suggest that the authors recheck it thoroughly. And possibly reconsider the whole plot style. It is quite difficult to follow. Also why are some labels horizontal and some vertical? Also the labels at bottom “Protein structure prediction” and “Small proteome annotation” are not making interpretation easier. I would describe the use cases in the manuscript but avoid having the labels in the chart.

Thank you. We agree on the complexity of Figure 2 and thank you for highlighting areas of improvement. We reworked the entire figure to improve clarity through the following points:

1. When numbers are bolded and horizontal, they indicate a baseline and its raw performance (also labelled in *ms* where appropriate). The remaining numbers, vertical in panel A and E and now including an “x” to designate relative

performance, i.e. the factor to the baseline, e.g. in panel A batch size 1, MMseqs2-GPU was **475ms**, while JackHMMER was **177.7x** slower than that.

2. We integrated the previous panel b cost improvement data into panel a. The number below each data point with the suffix “x\$” indicates the cost increase on cloud systems.
3. We now consistently indicate this through the orientation of the numbers across panels (a and d).
4. The above should also address the observation on JackHMMER between batch 1 and 100; while JackHMMER’s total execution is faster (thus plotted lower at 100 than 1), its relative performance to the baseline (MMseqs2-GPU) is slower in that setting.
5. We re-wrote the figure caption to highlight the meaning of horizontal and vertical number positioning.
6. We removed use-case labelling on the x axis for Panel A (i.e., “Protein structure prediction” and “Small proteome annotation”) to simplify readability, as indicated.
7. We merged the previous panels d and e into c to save horizontal space and highlight that MSA execution time was taking the majority of structure prediction time in AlphaFold’s case.

Supplementary data files: the “Supplementary Data 1 PDF” is empty.

The editorial platform appears to automatically create an empty PDF file with the name of the supplementary file when we attempt to upload a xlsx file as supplementary material. Unfortunately, we found no way to change this behaviour, thus for the Supplementary Data 1 xlsx there will still appear an empty PDF file.

Sensitivity benchmark: the discussion on sensitivity is limited to 2 paragraphs:

Sensitivity of the new GPU-based gapless algorithm (Pages 1, 2), the discussion only talks about estimation of sensitivity in order to benchmark performance against other methods (which is definitely needed). The data is available only in a table in one of the tabs in the supplementary spreadsheet.

Thank you. For the purposes of our analysis, i.e. a fair comparison across methods on speed, sensitivity is a parameter that we fix to be similar across methods, rather than a result. We have added a dedicated table listing sensitivity for the methods tested in Methods “Sensitivity” as an aid to the reader to avoid resorting to the Supplementary Data.

Sensitivity in the context of FoldSeek benchmarks (Page 3). A few numbers are given in the main text.

I think sensitivity analysis is essential to any new method like what’s proposed in this manuscript. The analysis deserves some more discussion and possibly a figure with

sensitivity benchmarks similar to that of previous mmseqs2 papers (can be in supplementary material). For instance, how about this question: is JackHMMER supposed to have essentially the same sensitivity as the GPU gapless method presented here (it looks like it does judging from the numbers, but it'd be nice to explicitly discuss whether there are/aren't conceptual differences in JackHMMER vs the GPU-gapless method introduced in this paper)

Thank you for this observation. Sensitivity is a tunable parameter in MMseqs2, Foldseek and BLAST. In the k-mer versions of MMseqs2 and Foldseek, the length of the generated similar k-mer list decides the speed-sensitivity trade-off (shorter k-mer lists yield faster execution at lower sensitivity). However, for methods like JackHMMER and MMseqs2-GPU, sensitivity is fixed due to their exhaustive prefilters. However, these tools can still reach higher sensitivity by using multiple *iterations* of the search (changing the workload from a sequence-sequence search, to a profile-sequence search). We have reworked the section to clarify this.

Indeed, MMseqs2 and JackHMMER are expected and perform similarly in terms of sensitivity in the initial sequence-sequence comparison. JackHMMER's profile-HMM search capability helps it achieve higher sensitivity in higher iteration counts. We added a sentence to clarify this:

[...] and slightly lower than JackHMMER, **which benefited from its Hidden-Markov-Model-based profiles in higher search iterations and reached** 0.614 and 0.685 at two and three iterations, respectively [...].

Reviewer #2

This manuscript introduces MMseqs2-GPU, a GPU-optimized version of MMseqs2 designed for protein sequence homology searches. MMseqs2 is widely adopted by the research community, making an accelerated version valuable.

We thank the reviewer for the encouragement and their comments.

However, several key limitations may hinder the widespread adoption of MMseqs2-GPU, as outlined below.

1. The speed benchmarks are based on a target database of 30 million sequences and 6,370 queries, which is significant but falls short of the scale of modern metagenomic datasets (e.g., billions of sequences).

We thank the reviewer for this observation. Indeed, it's difficult to pin down all the use cases alignment methods could be employed in. We tried to strike a balance by selecting two cases that best help in explaining our contribution, i.e. the first case is allowing for as fair as possible comparison between many tools, i.e. our previous MMseqs2 sensitivity benchmark with 6370 query- against 30M reference sequences, and the second, explored in the structure prediction use-case, of a few targets against a large reference database (including metagenomic derived ones). To better showcase MMseqs2-GPU's ability to deal with metagenomics-scale databases, we have expanded the explanations regarding the ColabFold-MMseqs2 search to highlight that it employs a large-scale three iteration profile search to search against two databases (UniRef100 and ColabFoldDB) with a total of 238M representatives and 1B cluster members.

In addition, we added Figure 1e to better present the performance of the gapless prefilter on real data and highlight that exceeding the amount of available GPU RAM results in only a modest performance impact (still running at ~63% speed of in-GPU-memory processing, see below).

Not all people have access to GPU. MMseqs2 may be the prior option as it is already fast at such database scale.

MMseqs2 on CPU is indeed efficient, allowing on top to trade-off between sensitivity and speed. It is worth highlighting that our benchmarks in Figure 1 and 2 are very generous towards CPU methods, leveraging a very capable CPU system with 128 cores/256 threads with 1TB of system memory and fast NVMe storage, generally out of reach for many academic research groups. However, when compared to more typical single-socket server systems with a more modest CPU thread count, MMseqs2-GPU becomes much more advantageous in its performance.

To highlight this, we compare MMseqs2-GPU to JackHMMER on a much more modest Google Colab Pro system with 6 CPU-cores, 64GB RAM and an L4 GPU with 24GB RAM using the same small query set containing 20 CASP14-FM sequences and the UniRef90 as a target database. The UniRef90 is sufficiently large (49GB) to exceed the amount of available GPU RAM on the L4 system. In this comparison using much more widely accessible hardware, we still observe a ten-fold speed increase over JackHMMER.

Thanks to the reviewers comment, we realized that our description of this benchmark could be improved. We have extended the paragraph in the main text to highlight the limited CPU hardware of the Colab Pro system and added a new methods section “Colab benchmark”.

The manuscript does not address how MMseqs2-GPU performs in extreme scenarios, where GPU memory and I/O bottlenecks could become critical limitations.

We thank the reviewer for raising this important point. Memory and I/O bottlenecks indeed represent fundamental challenges for both CPU- and GPU-based methods. We address these challenges through three strategies:

(1) Memory efficiency: MMseqs2-GPU reduces memory requirements compared to the CPU-based MMseqs2 k-mer index by storing only one byte per amino acid residue instead of approximately seven bytes per residue required in MMseqs2-CPU k-mer for its various data structures. This improvement enables efficient searching of large target databases, such as the two ColabFold databases that contain 238 million representatives and 1 billion cluster members, on a single NVIDIA L40S GPU. We now explain the improved memory efficiency of MMseqs2-GPU in the main text and note the size of the databases used in the structure prediction benchmarks in the Figure 2 caption, main text and methods section.

(2) Database streaming: MMseqs2-GPU efficiently handles databases that exceed GPU memory by streaming data from host RAM. This is now described in a new methods section “Database streaming.” Our benchmarks demonstrate that even when the database size is approximately four times larger than the available GPU memory, MMseqs2-GPU maintains roughly 60% of its peak performance. This efficiency is achieved through asynchronous CUDA streams, which overlap data transfers with GPU computations. We show this in a new Figure 1e and in the JackHMMER vs. MMseqs2-GPU Colab benchmark, both described above.

(3) Multi-GPU parallelization: For larger-scale searches, MMseqs2-GPU supports partitioning databases across multiple GPUs, enabling close to linear scalability in performance, provided the database fits into the combined GPU memory. We explain this briefly in a new methods section “Multi-GPU Parallelization.”

2. Can MMseqs2-GPU be used to search established metagenomic databases, such as the ColabFold database? It seems that the authors already did such experiment but without much details.

Yes, this is possible and was part of our protein structure prediction benchmarks. We have amended the manuscript to make it clearer that MMseqs2-GPU is indeed capable of conducting metagenomic-scale searches (see above).

How to search the ColabFold database with MMseqs2-GPU?

Generally, the MMseqs2-GPU backend is a drop-in replacement for MMseqs2 in ColabFold.

Users can additionally:

1. Re-index their databases to omit the k-mer data structures and thus lower memory consumption (see points above)
2. Start “GPU server” processes loading databases into memory in a permanent manner, allowing for more efficient execution (see new Methods section “GPU Server”).

We have added this documentation to the ColabFold instructions online. Additionally, we have clarified briefly the setup in Online Materials.

How to solve the issue of large memory requirement for big databases?

As long as databases fit in system memory, and given that MMseqs2-GPU requires less memory to represent the same starting database, database streaming remains efficient for large databases. See above for the description of the new Figure 1e. We chose not to add a benchmark for databases larger than the amount of available system RAM, as all tools become I/O bound at that point.

According to the manuscript, it seems that MMseqs2-GPU is just 3 times (GPU vs CPU!) faster than MMseqs2 for searching the ColabFold database. This may reduce the chance of being published in Nature Methods, as it is one of the widely used options in protein structure prediction.

Thank you for highlighting this point. First, we would like to highlight that since our initial submission, we have performed optimizations on both CPU and GPU execution of MMseqs2 for ColabFold searches, leading to different results reported in the new version of the manuscript and accompanying supplementary data sheets. In particular, ColabFold using MMseqs2-GPU is now 1.65x faster end-to-end compared to ColabFold using MMseqs2-CPU k-mer (was previously 3x), and 31.78x faster compared to AlphaFold (was previously 23.28x). ColabFold MMseqs2-GPU runtime improved from an

average of 18s to 11s by eliminating various overheads. The ColabFold MMseqs-CPU runtimes improved, also to a minor degree due to the previous point, however, mainly due to choosing optimized parameters. **Thus, the relative performance difference between the ColabFold pipelines has narrowed, while the performance of ColabFold pipelines vs. AlphaFold2 has widened.**

In general, the end-to-end performance increase switching to MMseqs2-GPU is *hurt* by the prediction execution time, which dominates the end-to-end execution time in ColabFold on GPU. To explain this better, it's useful to consider the data in Supplementary Data 1, "Folding tool comparison", reported here for convenience:

	AF2		ColabFold MMseqs2-kmer		Colabfold MMseqs2-GPU	
	JackHMMER +HHblits	AF2	MMseqs2 CPU k-mer	ColaFold	MMseqs2 GPU	ColabFold
Total (s)	39547.12	9072.147047	1204.49	1316.39	224.29	1305.58
per-item (s)	1977.36	453.61	60.22	65.82	11.21	65.28
per-item (~m)	33	8	1	1	0	1
Percent of total execution	81.34%	18.66%	47.78%	52.22%	14.66%	85.34%
Factor to MMseqs2-GPU	176.32		5.37		1	
Factor to MMseqs2 kmer	32.83		1			
Total per-item		2430.96		126.04		76.49
Factor to ColabFold MMseqs2-GPU		31.78		1.65		1
Factor to ColabFold MMseqs2 k-mer		19.29		1		

As is evidenced by the “Percent of total execution” row, between the AlphaFold2 pipeline and ColabFold using MMseqs2-GPU, the computational dependency is inverted, i.e. while in AlphaFold2 end-to-end the bottleneck is the MSA generation, in ColabFold using MMseqs2-GPU it’s the prediction of structure from the MSA. ColabFold using MMseqs2-CPU k-mer lies somewhere in between.

If we look at total folding runtime, **MMseqs2-GPU resulted 5.37x faster than MMseqs2 on CPU** to perform the ColabFold search, or 176x faster than AlphaFold2 with JackHMMER and HHBlits. This is a considerable difference, but more importantly inverting the computational bottleneck from MSA to structure prediction highlights how MMseqs2-GPU can unlock the community to focus on novel optimization work focusing on e.g. faster structure inference. We have expanded on this result in the main section of the manuscript.

Looking forward, we expect more protein language models to incorporate MSA and homology information. As summarized in a recent blog post by the lead author of ProteinGym (<https://pascalnotin.substack.com/p/have-we-hit-the-scaling-wall-for>), we see clear limits in pretrained-only pLMs, and many pLMs are now incorporating MSA

and structure. As pLMs inference has been much faster than structure inference, we expect MMseqs2-GPU to provide much larger accelerations for these types of models.

Lastly, achieving optimal performance for MMseqs2-CPU requires using either two separate systems, one for MSA generation with high CPU-core count and large amounts of RAM (at least 1TB to 2TB) and a separate GPU system for model inference, or alternatively, a single system that combines high CPU-core counts, large amounts of RAM and GPU(s). In the latter configuration, either CPU or GPU would be idle for a large fraction of the time. ColabFold with MMseqs2-GPU simplifies this by offering high speed on a single fully-utilized GPU-based hardware configuration.

3. The caption for Figure 2 states, “MMseqs2-GPU can split the target database across GPUs to enable larger database searches.” However, it is unclear how this splitting is implemented or achieved in practice.

We now briefly explain this in a new methods section “Multi-GPU Parallelization”.

4. The manuscript is difficult to follow at times. For instance, the term "TCUPS" appears in the Abstract without definition, which may confuse readers unfamiliar with its meaning.

We have rewritten the abstract to compare speedups in relative terms for the various use-cases that we show in the manuscript. We have also shortened the abstract to fit into the Brief Communication word limit. We have received a similar sentiment from the reviewers of the RECOMB-seq meeting that the manuscript’s methods were sparsely described. We have expanded the methods description to include seven new sections and considerably expanded five existing sections. In addition, we have carefully revised the main text and figures for clarity. We hope that these changes have made the manuscript easier to follow.

Additionally, the URL mmseqs.com redirects to GitHub. Why not directly provide the GitHub URL in the paper for clarity and convenience?

We chose to keep a link to a domain that we control to be able to migrate off GitHub if it becomes necessary at some point in the future.

5. The documentation for the GPU-optimized version is not user-friendly. For example, following the provided instructions led to an error message: “Database DB.fasta needs header information.” This suggests the instruction lacks clear guidance for users.

We thank the reviewer for carefully testing the software. We improved the README to include a brief quick start to MMseqs2-GPU and directly link to the wiki section that

describes how to use MMseqs2-GPU in detail. In addition, we have further expanded the wiki section “GPU-accelerated search” to avoid various pitfalls.

The error that the reviewer encountered stems from ``makepaddedseqdb`` expecting a MMseqs2 database created by ``createdb`` as input instead of a FASTA file. We have improved the documentation to make this clear.

Additionally, we also improved the README of ColabFold stating how to use MMseqs2-GPU for the MSA generation.